# Nitrate retention capacity of milldam-impacted legacy sediments and relict A horizon soils

Julie N. Weitzman[1], Jason P. Kaye[1]

[1]Department of Ecosystem Science and Management, The Pennsylvania State University, University Park, PA 16802, USA

*Correspondence to*: Julie N. Weitzman (jnw142@psu.edu)

**Abstract.** While eutrophication is often attributed to contemporary nutrient pollution, there is growing evidence that past practices, like the accumulation of legacy sediment behind historic milldams, are also important. Given their prevalence, there is a critical need to understand how N flows through, and is retained in, legacy sediments to improve predictions and management of N transport from uplands to streams in the context of climatic variability and land-use change. Our goal was to determine how nitrate ($NO_3^-$) is cycled through the soil of a legacy sediment strewn stream before and after soil drying. We extracted 10.16 cm radius intact soil columns that extended 30 cm into each of the three significant soil horizons at Big Spring Run (BSR) in Lancaster, Pennsylvania: surface legacy sediment characterized by a newly developing mineral A horizon soil, mid-layer legacy sediment consisting of mineral B horizon soil, and a dark, organic-rich, buried relict A horizon soil. Columns were first pre-incubated at field capacity, and then isotopically labeled nitrate ($^{15}NO_3^-$) was added and allowed to drain to estimate retention. The columns were then air-dried and subsequently rewet with N-free water and allowed to drain to quantify the drought-induced loss of $^{15}NO_3^-$ from the different horizons. We found the highest initial $^{15}N$ retention in the mid-layer legacy sediment (17±4%) and buried relict A soil (14±3%) horizons, with significantly lower retention in the surface legacy sediment (6±1%) horizon. As expected, rewetting dry soil resulted in $^{15}N$ losses in all horizons, with the greatest losses in the buried relict A horizon soil, followed by the mid-layer legacy sediment and surface legacy sediment horizons, respectively. The $^{15}N$ remaining in the soil following the post-drought leaching was highest in the mid-layer legacy sediment, intermediate in the surface legacy sediment, and lowest in the buried relict A horizon soil. Fluctuations in the water table at BSR which affect saturation of the buried relict A horizon soil could lead to great loses of $NO_3^-$ from the soil, while vertical flow through the legacy sediment-rich soil profile that originates in the surface has the potential to retain more $NO_3^-$. Restoration that seeks to reconnect the groundwater and surface water, which will decrease the number of drying-rewetting events imposed on the relict A horizon soils, could initially lead to increased losses of $NO_3^-$ to nearby stream waters.

**Keywords**: Legacy sediments; buried A horizon soil; $^{15}$nitrogen; tracer experiment

## 1. Introduction

Anthropogenic alterations of the landscape have impacted geomorphology and hydrology, with a major new development in eutrophication research recognizing that past land practices play an important role in the contemporary transfer of nutrients across landscapes (Renwick et al., 2005; Walter and Merritts, 2008a; Brush, 2009; Sharpley et al., 2013; Weitzman et al., 2014). Land manipulation has been a staple of agricultural societies, especially since the introduction of ploughing during the Neolithic agricultural revolution (~7500 years ago) (Hoffmann et al., 2007), followed by the expansion of agricultural land into steeper, forested upland areas during the Medieval period (~1300 years ago) (Williams, 2000) due to increases in human settlement populations (Larsen et al., 2016). Such land-use practices, in combination with the existence of tens of thousands of milldams throughout Europe by the 18$^{th}$ century (Downward and Skinner, 2005; Walter and Merritts, 2008a; Bishop and Muñoz-Salinas, 2013), dramatically altered the European landscape through increased soil erosion and sediment redistribution (De Brue and Verstraeten, 2014; Larsen et al., 2016).

Similarly, serving as a case study for the research of this study, the historic, post-European settlement of the mid-Atlantic region in the U.S. in the seventeenth and eighteenth centuries was characterized by rapid anthropogenic landscape modifications, consisting of intense land clearing, deforestation, and the construction of tens of thousands of milldams (Walter and Merritts, 2008a; Merritts et al., 2011). Pervasive land clearing led to increased sedimentation rates throughout the Chesapeake Bay watershed (Jacobson and Coleman, 1986; Brush, 2009), with much of this sediment being deposited and stored behind small (~2.5-3.7 m high), valley-spanning milldams (Walter and Merritts, 2008a). Following abandonment in the nineteenth and twentieth centuries, many of these small milldams breached, which led to stream incision through the previously impounded sediment column. Subsequent lateral stream propagation can then lead to lowering of the water table, eventually exposing the former mill pond sediments as a new valley bottom terrace. This new, post-settlement sediment that overlies the original valley bottom is often referred to as legacy sediment.

Post-settlement modification of the land, typified by accelerated upland erosion and the ubiquitous construction of milldams, has increased both the rate of sediment input and the number of sediment sinks within the Piedmont region, greatly changing the area's fluvial geomorphology (Renwick et al., 2005; Walter and Merritts, 2008a). The deposition of fine-grained legacy sediment throughout the mid-Atlantic region, in particular, has led to the burial of once biogeochemically active riparian valley bottoms (Merritts et al., 2005; Walter and Merritts, 2008a; Merritts et al., 2011), which, in turn, has altered nutrient cycling dynamics at the land-streamwater interface (Meade et al., 1990; Renwick et al., 2005; Walter et al., 2007; Walter and Merritts, 2008a; Merritts et al., 2011; Weitzman et al., 2014). Legacy sediments introduce two key problems for water quality. Firstly, erosion of deeply incised, fine-grained stream banks is a significant non-point source of suspended sediment and nutrients entrained in the sediment, which can contribute to contemporary nutrient loading to downstream waterways (Trimble, 1997; Walter and Merritts, 2008a; Gellis et al., 2009; Gellis and Mukundan, 2013). In the mid-Atlantic, legacy sediments

constitute a substantial volume of sediment stored in stream corridors (Banks et al., 2010; Massoudieh et al., 2013; Gellis and Noe, 2013). Lancaster County, in particular, is recognized as a hotspot for high sediment and nutrient yields to the Chesapeake Bay, with bank erosion of legacy sediments acting as a major source of these pollutants (Merritts and Walter, 2003). Secondly, legacy sediment-dominated stream banks alter flowpaths for water and dissolved nutrients (Walter and Merritts, 2008a), affecting present-day nutrient transfers from uplands to streams. The incised, high-banked channels characteristic of legacy sediment-strewn streams are interpreted to be fill terraces, as opposed to floodplains (Walter and Merritts, 2008a), resulting in the distinct physical separation of biogeochemically active zones (surface soils) from subsurface hydrologic flowpaths. Seasonal drying-rewetting events, largely controlled by fluctuating water tables, are common in legacy sediment-impacted streams, and have the potential to impact the release or retention of nitrogen (N). Given their prevalence throughout the mid-Atlantic region, there is a critical need to understand how N flows through legacy sediments in order to improve predictions and management of N transport from uplands to streams.

Floodplains are known to be active N sinks that can support high N retention in sediments of adjacent water bodies (Forshay and Stanley, 2005; Kaushal et al., 2008b; Harrison et al., 2011). In contrast, legacy sediment-rich fill terraces have been shown to dampen N removal pathways in the long-buried relict soils which they overlie, while also acting as potential sources of nitrate ($NO_3^-$) to waterways (Weitzman et al., 2014). Climate-driven export of N from watersheds is known to occur (Howarth et al., 2006; Lewis and Grimm, 2007; Kaushal et al., 2008a; Kaushal et al., 2010; Duncan et al., 2015), with N stored during “dry” years or seasons and flushed from watersheds during “wet” years or seasons. Such drying-rewetting cycles have been linked to region-wide pulses of high $NO_3^-$ concentrations in tributaries of the Chesapeake Bay (Kaushal et al., 2010). Climate change models predict an increase in the variability of precipitation and hydrologic events, as well as an increase in the intensity of extreme weather conditions (IPCC, 2013). Specifically, in the mid-Atlantic region, recent records show increasing rainfall intensities that coincide with the hurricane season in October (Spierre and Wake, 2010; Lu et al., 2015), with climate change models forecasting drier autumns and wetter winters for the region (Shortle et al., 2015). In addition to these more extreme precipitation events, longer periods with no precipitation are expected, with droughts predicted to become worse during the mid-Atlantic summers (Hayhoe et al., 2007). During such extreme weather events nutrient retention versus release will impact whether large pulses of $NO_3^-$ are flushed from landscapes to streams in the future (Kaushal et al., 2010).

While prior research has identified the post-drought $NO_3^-$ pulse in mid-Atlantic streams (Kaushal et al., 2010), the mechanisms that lead to such $NO_3^-$ pulses have not yet been characterized. Previous work at BSR suggests that $NO_3^-$ created in surface soils on legacy sediment terraces may not be effectively removed if transported through buried relict soils, with legacy sediments potentially acting as an important source of $NO_3^-$ that can be flushed into nearby streams (Weitzman et al., 2014). However, it is unclear whether carbon-rich relict soils that are exposed will act as a source or a sink for $NO_3^-$, especially when further impacted by drought conditions. As such, our study aims to understand how legacy sediments influence the

transfer of $NO_3^-$ from soils to streams by contrasting the dominant geomorphic (soil horizons), climatic (drought), and cultural (restoration) sources of variation in $NO_3^-$ retention capacity in legacy sediment strewn streams. Specifically, we focused on the following three questions: (1) Geomorphic: In which soil horizon (surface legacy sediment, mid-layer legacy sediment, or relict A horizon soil) will initial $NO_3^-$ retention be greatest? (2) Climatic: Which soil horizon will experience the largest drought-induced $NO_3^-$ flush following sequential leaching? (3) Cultural: Will restoration (i.e. exposure of the bottom horizon) alter losses in the relict A horizon soil? We answered these questions by tracing the fate of $^{15}NO_3^-$ through intact soil cores extracted from the Big Spring Run watershed in Lancaster, Pennsylvania.

Decades of research assessing the role of near-stream ecosystem function have been conducted (e.g. Hill, 1996; Carpenter et al., 1998) without considering how such zones were modified by legacy sediment deposition as a consequence of dam building and breaching. Quantifying changes in N retention among the soil horizons typical of legacy sediment terraces will provide critical information for assessing sources of N to streams, improving the efficacy of riparian buffers on legacy sediments, and understanding the effects of past land use on contemporary N flow from soils to streams.

## 2. Materials and Methods

### 2.1 Field Site

Big Spring Run (BSR) (39°59' N, 76°15' W) is a northward-flowing tributary of Mill Creek in West Lampeter Township, in central Lancaster County, Pennsylvania (drainage area ~4 km$^2$). The BSR Watershed is a sub-basin of the Conestoga River Watershed, which, itself, empties into the Lower Susquehanna River. The Susquehanna River eventually flows into the Chesapeake Bay, and provides >50% of the freshwater, and, as of 2009, 46%, 26%, and 33% of the total N, phosphorous (P), and sediment, respectively, delivered to the Bay (Chang 2003; USEPA, 2010; PA DEP, 2011). BSR is typical of many headwater watersheds in the temperate climate Piedmont Physiographic Province, which are characterized by low valley slopes (~0.005) and relief (~30 m) (PA DEP, 2013). Soils in the BSR Watershed consist of deep, silty loams derived from Conestoga limestone (Merritts et al., 2005). The somewhat poorly drained Newark soil series (Fluventic Endoaquepts) predominates near the legacy sediment strewn stream and gradually grades into the well-drained Pequea soil series (Typic Eutrudepts) in the uplands (Custer, 1985). Soils utilized in this study were only of the Newark soil series. A typical Newark profile includes an A horizon (Ap: 0-23 cm) underlain by B (Bw: 23-38 cm; Bg: 38-81 cm) and C (Cg: 81–152 cm) horizons.

As in many stream banks of the mid-Atlantic Piedmont region impacted by legacy sediment deposition, the soils along BSR consist of four principle stratigraphic units (see Fig. 1), which from bottom to top include: (1) Pleistocene periglacial basal gravels that overlay bedrock; (2) buried A horizon soils that are a relict of pre-settlement, Holocene soil development; (3) legacy sediment deposits (post-settlement alluvium and colluvium) that buried the relict A horizons; and (4) newly formed

A horizon soils that developed *in situ* on legacy sediment terraces as they were converted to agricultural "bottom lands" for crops and/or grazing. While previous work classified the long-buried A horizons as relict hydric soils, we find there is still conflicting evidence as to whether the soils can be distinctly delineated as hydric. As such, we have decided to take a more conservative approach, and classify the buried soils as relict A horizons.

The basal gravels are composed of angular to subangular quartz cobbles that directly overlie bedrock of Conestoga limestone. Evidence suggests that these poorly sorted gravels are derived from Pleistocene periglacial lag deposits that served to concentrate and direct shallow groundwater flow in the valley bottom (PA DEP, 2013). The construction of numerous, small beaver dams during pre-settlement times (Morgan, 1867; Walter and Merritts, 2008b; Brush, 2008), in addition to the flow conditions created by the periglacial lag deposits, likely led to the development of a fluvial wet meadow environment over the last 10,000 years, during the Holocene (Merritts, et al., 2005; Walter and Merritts, 2008a; Merritts et al., 2011). Remnants of this pre-settlement, Holocene valley bottom currently exist at BSR as dark (10 YR 2/1), fine-grained (loam), organic matter-rich, 20-50 cm thick relict A horizon soils above the basal gravels. Similar valley bottoms characterized by shallow anabranching channels flowing through islands of low vegetated wet meadows were once pervasive throughout the mid-Atlantic Piedmont region (Walter et al., 2007; Walter and Merritts, 2008b; Merritts et al., 2011). These pre-settlement wet meadows stored large amounts of organic-rich material, but little sediment due to the low, long-term erosion rates in pre-settlement times and frequent overbank flow onto the broad, riparian floodplains (Walter et al., 2007; Walter and Merritts, 2008b).

Accelerated soil erosion due to post-settlement practices coincided with the construction of numerous milldams in the mid-Atlantic, with such dams typically spanning entire valley bottoms of dominantly 1$^{st}$ to 3$^{rd}$ order streams, rising to heights that averaged 2.5 m (Walter and Merritts, 2008a). These dams created long, linear millpond reservoirs that flooded the once extensive wet meadow valley bottoms several kilometers upstream, eventually becoming efficient sediment retention ponds. The uniform, fine-grain size of such post-settlement legacy sediments (dominantly silt-clay, with massive, occasionally horizontal bedding) suggests they were deposited in very low velocity waters, characteristic of slack-water environments (PA DEP, 2006). Such conditions argue against the idea that legacy sediments were deposited as floodplains, as none of the characterizations of floodplain deposits were observed (i.e. fining-upbed grain deposits, etc.). Furthermore, pollen analysis of pre- and post-settlement deposits at BSR show vegetation consistent with through-flowing water conditions during pre-settlement times versus stagnant, slough-like conditions in post-settlement times (Merritts et al., 2005; Voli et al., 2009).

At BSR a 3 m high milldam once existed about 2 km downstream from its headwaters. During the historic, post-settlement period, legacy sediments (~80-100 cm thick) were deposited on top of the A horizon soil horizon behind this former milldam. Prior to restoration in September 2011, a gradient of legacy sediment depth existed at BSR, with sediments thickest near the former milldam, and tapering off upstream away from the dam. Following dam breachment at BSR in the early 20$^{th}$

century deep channel incision into the stored millpond sediment led to the formation of high-banked channels, exposing the post-settlement legacy sediment, buried relict A horizon soil, periglacial basal gravels, and underlying valley bedrock, and effectively eliminating hyporheic exchange between surface water and groundwater (Water and Merritts, 2008a; Merritts et al., 2011; Parola and Hansen, 2011). A new surface A horizon, ~20 cm thick, has since developed on top of the historic legacy sediment.

In September 2011 The Pennsylvania Department of Environmental Protection (PA DEP) designated a portion of BSR as a test site for implementing and monitoring a new Best Management Practice (BMP) that was specifically targeted to streams in the eastern US impacted by damming (Hartranft et al., 2011). This natural aquatic ecosystem restoration design seeks to re-establish the natural function and condition of the stream, floodplain, and riparian zones within BSR (PA DEP, 2009). To reconnect the original floodplain hydrology of the site, legacy sediment was removed throughout a segment of the BSR watershed, exposing the once-buried relict A horizon soil. Legacy sediment accumulation, and its controlled removal from a portion of the watershed, has been extensively mapped at BSR, and, as such, made it the ideal test site for our objectives.

### 2.2 Soil Column Sampling and Preparation

In June 2011 we extracted 5 replicate intact soil columns that extended 30 cm into each of the three significant soil horizons at BSR (surface legacy sediment, mid-layer legacy sediment, and relict A horizon soil), resulting in a total of 15 separate intact soil columns. The original sampling scheme which sought to collect intact soil columns that extended through all three soil horizons was abandoned due to excessive compaction when cores were >100 cm. Instead, ~30 cm long Schedule 80 PVC pipe (20.32 cm inner diameter) was pushed into the soil by a 2 Mg drop weight that was slowly lowered onto the upright PVC pipe. Surface legacy sediment cores were collected at a depth of 0-30 cm, mid-layer legacy sediment cores from a 45-75 cm depth, and buried relict A horizon soils were collected at a depth of 105-135 cm. A step-wise sampling scheme was created in order to sample each of the three significant soil horizons with minimal boundary effects (Fig. 1). A backhoe was used to remove the surface legacy sediment from one section (to sample the underlying legacy sediment), and was used again to remove the surface and mid-layer legacy sediment from another area (to sample the underlying relict A horizon soil). The PVC pipes were inserted into the soil using the drop weight. Once the pipes were in the ground surrounding sediment was removed and the pipes were tilted to cleanly break contact between the soil column and the underlying subsoil. Each column was then inverted and washed sand was poured into voids created by the separation of the soil at the column bottom. The sand was covered by nylon drain fabric and then a PVC disk was inserted and held in place by a PVC cap. Each PVC cap was outfitted with an outlet port to allow for leachate collection. The intact soil columns were transported to Penn State University, where they were maintained in the upright position at field soil moisture. Soil moisture and temperature were monitored continuously throughout the project using Decagon 5TM soil moisture and temperature sensors (Decagon Devices, Inc. Pullman, WA)

inserted vertically in the top 5 cm of soil in each core. Mean volumetric soil moisture when the cores were collected was 33% for the surface legacy sediment horizons, 23% for the mid-layer legacy sediment horizons, and 38% for the relict A horizon soils.

Each soil column was brought to its respective field capacity by saturating from above with nitrogen-free water and leaving them to drain for three days. To ensure that each soil horizon was sufficiently saturated, each core was flushed with a total of 12 L of 0.001 M potassium sulfate ($K_2SO_4$) solution – enough to fill all soil pores in the respective soil horizon twice. The addition of such a large amount of solution caused some ponding on the column surfaces. As the cores freely drained leachate was collected from the outlet ports at the bottom of each core, and frozen at 4˚C for future analysis.

**2.3 Soil Nitrogen Retention Experiments**

Once all soil columns reached field capacity (within 3 days after the lab saturation) an isotopically-enriched solution of nitrate ($^{15}NO_3^-$, 60% APE or atom percentage excess) was added to the surface of each column and allowed to drain freely out of the lowest portion of the column. A total of 12 L of solution, which amounted to double the pore space of each soil horizon, was steadily added to the surface of each core in order to ensure complete saturation. This scheme resulted in periods of ponded water, but all solution eventually drained through the cores. Leachate was collected from all columns for $^{15}NO_3^-$ analysis, as described below. The isotope-spiked solution added to each soil column contained 5.36 mg $NO_3^-$-N $L^{-1}$ (or about 2 g N $m^{-2}$) made from 60% APE $Ca(^{15}NO_3)_2$. We selected this trace concentration because our experience throughout agricultural landscapes of Pennsylvania, and Big Spring Run in particular (Weitzman et al., 2014), suggested that this was a typical value for soil water. Using this combination of nitrogen (N) concentration and $^{15}N$ enrichment also ensured that we had added enough $^{15}N$ to label soil and water at detectible levels. When water no longer dripped from the soil columns a subsample of soil was taken from each soil column (from the soil surface to the PVC disk; 30 cm depth total) using a 2 cm diameter soil probe. A PVC pipe (2 cm x 40 cm) was placed in each column to fill the space created by the sample coring to minimize the alteration of water infiltration and percolation through the soil. Cores were left to dry for ~1 month, reaching a steady dry volumetric soil moisture content of < 15 %. This moisture content represents soils that are nearly air-dry (though the deeper relict A horizon soils were not as dry at ~17 %), mimicking field drought conditions. Lastly, 12 L of N-free water (0.001 M $K_2SO_4$) was steadily added to each "dry" column and the leaching and soil subsampling procedures were repeated.

**2.4 Soil Analysis**

All soil samples were homogenized by hand prior to subsampling for specific analyses. Soil inorganic N concentrations (µg N g $soil^{-1}$) were determined on soil subsamples by extracting with potassium chloride (100 mL of 2.0 M KCl) and analyzing via colorimetric analysis on a spectrophotometer microplate reader. Ammonium ($NH_4^+$) concentrations were measured using the

salicylate method (Sims et al., 1995), while nitrate ($NO_3^-$) + nitrite ($NO_2^-$) concentrations were determined using the vanadium (III) chloride method (Doane and Horwath, 2003). Concentrations of $NO_2^-$ were assumed to be negligible, so results are reported only as $NO_3^-$-N concentrations.

Gravimetric water content for each soil was determined by oven drying (105 ˚C) a separate 10 g sieved (2 mm) subsample to constant mass. Dried subsamples of soil were then ground on a roller mill, rolled in tin capsules, and analysed by dry combustion elemental analysis followed by isotope ratio mass spectrometry at the Boston University Stable Isotope Laboratory to determine the concentration and isotopic ratio of N within each soil horizon, as well as total soil carbon (C). The fraction of added $^{15}N$ that was retained in the different soil horizons both pre-drought and post-drought were calculated using the following standard mixing equations (Kaye et al., 2002a, b):

$$N_0 = N_a + N_n \quad (1)$$

$$N_0{*}^{15}N_0 = N_a{*}^{15}N_a + N_n{*}^{15}N_n \quad (2)$$

Rearranging Eq. (1), and substituting into Eq. (2):

$$N_a = (N_0{*}^{15}N_0 - N_0{*}^{15}N_n)/(^{15}N_a - {}^{15}N_n) \quad (3)$$

Where $N_0$ is the total mass of N in the soil, $N_a$ is the mass of added tracer N in the soil pool, $N_n$ is the mass of native soil N, $^{15}N_0$ is the atom % $^{15}N$ enrichment in the soil sample, $^{15}N_n$ is the atom % $^{15}N$ enrichment of the native soil N, and $^{15}N_a$ is the atom % $^{15}N$ enrichment of the added tracer N.

**2.5 Leachate Analysis**

Leachates from each treatment were analyzed colorimetrically, as described above, to quantify $NH_4^+$ and $NO_3^-$ concentrations. A bulk leachate sample was taken for each column and solution treatment following draining. A time series of leachate samples was also taken following the addition of the $^{15}NO_3^-$ enriched solution from one column of each of the soil horizons of interest to determine release of $^{15}NO_3^-$ over time. All leachate samples were sent to the U.C. Davis Stable Isotope Facility for $^{15}NO_3^-$ analysis in water using the denitrifier method (Sigman et al., 2001; Casciotti et al., 2002).

The fraction of added $^{15}NO_3^-$ that passed through each soil horizon in the leached solution was then calculated from the same standard mixing model principles (Kaye et al., 2002a, b) as detailed in Eq. 1-3, but where $N_0$ is the total mass of N as $NO_3^-$ in the bulk leachate, $N_a$ is the mass of added tracer N, $N_n$ is the mass of native leachate N as $NO_3^-$, $^{15}N_0$ is the atom % $^{15}N$ enrichment in the bulk leachate $NO_3^-$, $^{15}N_n$ is the atom % $^{15}N$ enrichment of the native leachate $NO_3^-$, and $^{15}N_a$ is the atom % $^{15}N$ enrichment of the added tracer N. Similarly, the amount of $^{15}NO_3^-$ remaining in the pore water of each soil column was calculated, with all terms in the standard mixing model the same as those used to calculate $^{15}NO_3^-$ in leachate, expect for $N_0$, which in this case is the total mass of N as $NO_3^-$ in the remaining pore water of the soil columns.

### 2.6 $^{15}$N Recovery vs. Retention

Total $^{15}$N *recovered* was calculated as the sum of $^{15}$N retained in the soil plus $^{15}$N measured in the leachate and $^{15}$N in the pore water immediately following the addition of the isotope-labeled solution to the field-capacity soil columns (i.e. pre-drought), as well as after rewetting and draining the cores (i.e. post-drought). *Retained* $^{15}$N was calculated as the mass of tracer $^{15}$N found remaining in the soil and pore water both pre- and post-drought.

The pulsed loss of $^{15}$N following the subsequent rewetting of the dried soil columns was also calculated for each soil horizon as the percent decrease in $^{15}$N retention between the two moisture conditions, as shown in equation 4:

$$^{15}\text{N Pulsed Loss From Soil} = [(^{15}\text{N}_{PRE} - {}^{15}\text{N}_{POST})/(^{15}\text{N}_{PRE})] \times 100 \quad (4)$$

Where $^{15}N_{PRE}$ is the mass of $^{15}$N retained in the soil and pore water pools pre-drought and $^{15}N_{POST}$ is the $^{15}$N retained in the soil and pore water pools post-drought.

### 2.7 Soil Properties

Soil texture and particle-size distribution were determined for each of the three soil horizons of interest by analysing subsamples of soil from each intact soil column according to Kettler et al. (2001). This rapid, simplified method for evaluating particle-size distribution employs a combination of sieving and sedimentation steps. A 3% aqueous concentration of sodium hexametaphosphate [HMP, $(NaPO_3)_n$] was added to soil samples (<2 mm) in a 3:1 HMP to soil ratio and placed on a shaker for 2 hours to allow for dispersion of individual soil particles and to aid in the break-down of soil aggregates. Following this dispersal step, sieving and sedimentation procedures were used to fractionate the soil particles into sand, silt, and clay size classes.

Unsaturated hydraulic conductivity was also estimated for each soil horizon by experimentally determining soil water retention curve measurements in the laboratory via pressure-plate extraction as described by Dane and Hopmans (2002b) and then fitting the van Genuchten soil hydraulic model (van Genuchten, 1980) to the experimental data. Soil hydraulic parameters for the van Genuchten model were estimated using the SWRC Fit program (Seki, 2007), while saturated hydraulic conductivity values for the model were determined based on USDA soil texture classes and porosity calculated from bulk density measurements (Rawls et al., 1998). Undisturbed soil samples (enclosed in metal rings of 5 cm diameter and 2.5 cm height) were collected from the intact soil columns following the drying-rewetting experiment. The soil samples, retained in the metal rings, were placed on wet porous ceramic plates and equilibrated at 6, 10, 33, 100, and 200 kPa in a pressure-plate extractor (Soilmoisture Equipment Corp., Santa Barbara, CA). Two replicates were used for each soil layer and pressure step.

### 2.8 Statistical Analyses

Repeated-measures analysis of variance (ANOVA) was carried out in Minitab 17.1 (Minitab Inc., State College, PA) to examine the differences in measured properties, including $^{15}N$ retained in the soil, $^{15}N$ leached through the soil, $^{15}N$ remaining in the pore water of the soil, total soil C and N, soil extractable $NO_3^-$ and $NH_4^+$, leached $NO_3^-$ and $NH_4^+$ through the soil, total volume of leachate, and $\delta^{15}N$ with time (pre-drought and post-drought, with field capacity included for only some properties), and across the three main soil horizons of BSR (surface legacy sediment, mid-layer legacy sediment, and relict A horizon soil). All data were checked for normality, homoscedasticity, and outliers, and transformed when necessary prior to carrying out ANOVA analyses. Specifically, data corresponding to $^{15}N$ retained in the soil pre-drought was natural log transformed, as were pre-drought and post-drought total C and N data, pre-drought soil leached $NO_3^-$, and pre-drought and post-drought leachate volume drained. Soil horizon, time of sampling (pre-drought or post-drought), and the interaction term between the two were used as factors in the repeated-measures ANOVA model. When main effects or interactions were found to be significant ($\alpha$ = 0.05), data were further analysed by a one-way ANOVA, and a Tukey's HSD post-hoc test (with 95% confidence limits) was used to compare differences across the specific soil horizons and/or sampling conditions. A paired t-test was used to compare both $^{15}N$ retention and $^{15}N$ leaching values pre-drought versus post-drought for each of the three soil horizons of interest at BSR, to determine if pulsed losses of $NO_3^-$ were significant. Since the number of sample pairs in each comparison was low (<30), differences were checked for normal distribution. Bulk density, porosity, and particle size distribution classes (sand, silt, and clay) were analysed using standard one-way ANOVAs across the three soil horizons.

## 3. Results and Discussion

### 3.1 Soil Nitrogen Dynamics

Total recovery of applied $^{15}N$ (sum of soil, leachate, and pore water $^{15}N$ pools, Fig. 2) in pre-drought soils was significantly different among the three soil horizons ($P < 0.01$), with a total recovery of 63% in the surface legacy sediment horizon that was significantly lower than the total recovery of 92% in the mid-layer legacy sediment horizon and 96% in the relict A soil horizon. While extraction efficiencies are never 100% due to unexplained abiotic and biotic processes that can rapidly consume $^{15}N$ immediately following isotope addition (Davidson et al., 1991; Hart et al., 1994), extraction efficiencies for $^{15}NO_3^-$ typically range from 90-95% (Norton and Stark, 2011). The lower total recovery of tracer $^{15}N$ in the surface legacy sediment horizon of BSR is likely due to a number processes, including possible gaseous losses of $^{15}NO_3^-$ via denitrification or nitrification (Morier et al., 2008; Templer et al., 2012), or translocation of N through plant roots or fungal hyphae (Rütting et al., 2011).

Retention of applied $^{15}N$ tracer in the soil and pore water pools combined was not significantly different among soil horizons ($P > 0.05$) for either pre- or post-drought conditions (Fig. 3). However, when the two pools were analyzed separately, $^{15}N$ retention in the soil pool was found to be statistically significant for the three soil horizons both pre- and post-drought, while $^{15}N$ retention in the pore water pool was not found to be statistically significant across soil horizons or moisture conditions (Fig. 3). In field-capacity soils, pre-drought, both the mid-layer legacy sediment and relict A horizon soils retained a significantly higher percent of added $^{15}NO_3^-$ than the surface legacy sediment horizon. Overall, the mid-layer legacy sediment horizon had the highest initial (i.e. pre-drought) soil $^{15}N$ retention at 17%, followed by the relict A horizon soil with 14%, and lastly the surface legacy sediment horizon with 6% retention. A paired t-test revealed that soil $^{15}N$ retention was significantly lower ($P < 0.01$) post-drought, as compared to the initial field-capacity soils, pre-drought, with a decline in soil $^{15}N$ retention in all three soil horizons over this time. Though mid-layer legacy sediments showed the greatest soil $^{15}N$ retention post-drought, it was a fairly small amount, with only 3% of applied $^{15}N$ tracer still being held in the soil. Surface legacy sediments had a similar soil $^{15}N$ retention capacity of 2% post-drought, while relict A horizon soils retained almost none of the $^{15}N$ that it initially held - only 0.8% of the originally applied tracer amount. The decline in $^{15}N$ retained in the soil and pore water pools combined following the sequential leaching event post-drought represents a pulsed loss of $NO_3^-$ from the soil. This loss was largest in the relict A horizon soil, where 90% of the $^{15}N$ tracer initially held in the soil columns was released upon rewetting of the dry soil. The surface and mid-layer legacy sediment horizons had large pulsed losses of $^{15}N$ as well, releasing 69% and 59% of the originally retained tracer, respectively.

Losses of $^{15}N$ as leachate significantly exceeded the amount of $^{15}N$ that was retained in the soil and pore water in all three horizons at BSR ($P < 0.01$) (Fig. 2). The mid-layer legacy sediment and relict A soil horizons had statistically similar leaching losses of $^{15}N$ pre-drought, with 66 and 69% lost, respectively. While $^{15}N$ lost as leachate was found to be lower in the surface legacy sediment horizon, as compared to the other two horizons, it was still a large amount at 50%. Release of $^{15}NO_3^-$ versus native $NO_3^-$ over time in pre-drought leachate for one core for each of the three soil horizons showed the expected isotope-mixing trend of higher $^{15}NO_3^-$ values (atm%; Fig. 4a) corresponding with lower native $NO_3^-$ values (mg $L^{-1}$), and vice-a-versa. Though only one core per soil horizon was analyzed for $^{15}NO_3^-$ release over time, we interpret and discuss the release curves as being representative of each of the three soil horizons at BSR given that the replicate cores for each soil horizon of interest showed similar $NO_3^-$ leaching trends over time (Fig. S1 in the Supplement). Early in the leaching nearly all of the N leached from surface and mid-layer legacy sediment horizons was native N (very low atm%), while the relict A horizon soil had about 2/3 native N and 1/3 added N. Later in the leaching event, almost all of the N in leached solution had come from the added $^{15}NO_3^-$, as indicated by the atm% near 60 (i.e. the enrichment level of the added $^{15}NO_3^-$ tracer). The high concentrations of native $NO_3^-$ that are not immediately replaced by $^{15}NO_3^-$ in the surface legacy sediment horizon of BSR suggests that $NO_3^-$ may be sitting in the water column (as is confirmed by the $^{15}N$ in pore water data), and thus high concentrations may be easily

flushed from the surface legacy sediment horizon after small precipitation events. Only as the levels of native $NO_3^-$ decrease does $^{15}NO_3^-$ begin to be released from the surface legacy sediments, indicating that larger and longer precipitation events (i.e. saturated conditions) are needed in order to release any newly added $NO_3^-$ inputs. The early detection of $^{15}NO_3^-$ in the leachate from the relict A horizon soils, on the other hand, may reflect some influence of preferential flow, either natural or created during coring and column construction. If the pattern is representative of the relict A horizon soil and not an artifact of laboratory conditions, then early detection of $^{15}NO_3^-$ suggests that some $NO_3^-$ added to the relict A soil horizon may be quickly lost due to short interaction time with soil particles. Initial $NO_3^-$ concentrations leached from the mid-layer legacy sediment and relict A horizon soils were much lower than in the surface legacy sediment horizon (Fig. 4b), which is in agreement with the initially higher soil retention rates found for these two horizons (Fig. 3).

Post-drought, $^{15}N$ leachate losses in the three soil horizons were lower than found pre-drought, as shown by a paired t-test ($P < 0.05$). This is not surprising, however, given that >50% of the originally applied $^{15}N$ had already been leached from the soils before the drought treatment was imposed. The leached $^{15}N$ pool accounted for 77% of the pulsed loss of $^{15}N$ from the relict A horizon soil, post-drought. Such leaching losses of $^{15}N$, however, could only account for 50% and 44% of the pulsed $^{15}N$ loss that occurred in the surface and mid-layer legacy sediment horizons, respectively. We cannot rule out that the remaining portion of lost $^{15}N$ was released from the soil via dissolved organic N desorption or gaseous losses that were not measured. Nor can we quantitatively assess whether the $^{15}NO_3^-$ measured in the leachate solutions passed straight through the soil core systems untransformed, or whether the $^{15}NO_3^-$ underwent rapid assimilation and remineralization. If $^{15}NO_3^-$ is assimilated to an organic form, and this $^{15}N$-labeled organic form is then remineralized, nitrified, and flushed from the system, it will impart the same $^{15}NO_3^-$ signature as untransformed $^{15}NO_3^-$ (Curtis et al., 2011). This assumes that little of the $^{15}N$ label was retained in more stable pools of organic N, which is consistent with our results. However, past work has shown that microbial activity, especially in the mid-layer legacy sediment and relict A soil horizons, is low (Weitzman et al., 2014). This suggests that $NO_3^-$ assimilation/remineralization processing is not likely occurring in these deeper soils at BSR, rather, newly deposited $NO_3^-$ is probably transported through the subhorizon soil system unaltered.

Losses of N can occur when inputs exceed the maximum net N sink size of the system (capacity N saturation), or when input rates exceed net retention rates (kinetic N saturation) (Lovett and Goodale, 2011). Leaching of N as $NO_3^-$, in particular, suggests that biological sinks for $NO_3^-$ are too small or too slow to prevent losses. There is evidence that $NO_3^-$ can be retained in soils over short time scales (seconds to minutes) (Davidson et al., 2003; Fitzhugh et al., 2003; Corre et al., 2007), with transformations between inorganic and organic N forms also occurring rapidly (hours to days) (Lewis et al., 2014; Weitzman and Kaye, 2016), even in soil systems with short hydrological retention times and low organic matter contents (Campbell et al., 2002; Wynn et al., 2007). The low initial $^{15}N$ retention in the soil for all three horizons of interest, especially in the surface legacy sediment horizon, which initially retained only 6% of the applied tracer, suggests that the soil may already

be near C and N saturation from a mineral protection perspective (Schmidt et al., 2011; Castellano et al., 2012). The increased direct loss of $NO_3^-$ via leaching in the three soil horizons may also be an indicator of N saturation. The relict A horizon soils had the largest C and N pools (Table 1), and if the mineral particles are close to saturation with organic matter, then coupled C and N saturation theory suggests that it could retain less N inputs (Castellano et al., 2012). Preferential uptake of $NH_4^+$ relative to $NO_3^-$ may also inhibit $NO_3^-$ uptake or immobilization (Rennenberg and Gessler, 1999; Bradley, 2001; Emmett, 2007). Concentrations of soil extractable $NH_4^+$ are much lower than concentrations of soil extractable $NO_3^-$ in all three soil horizons (Table 4), which could be due to greater uptake of $NH_4^+$. Furthermore, potential nitrification rates were previously found to be low for both the mid-layer legacy sediment and relict A soil horizons at BSR, likely due to the low potential activity of $NH_4^+$ oxidizer communities (Weitzman et al., 2014). Taken together these data suggest that low $NO_3^-$ uptake, rather than enhanced nitrification, may be responsible for the large $NO_3^-$ leaching losses measured in the two subsurface soil horizons at BSR.

### 3.2 Soil Properties

Water retention, which relates to a soil's ability to store water, and hydraulic conductivity, which is the measure of a soil's ability to transmit water, are the two main soil properties that determine the behaviour of a soil's water flow system (Klute and Dirksen, 1986). These two hydraulic properties are primarily dependent upon the particle-size distribution of the soil, and the structure of these particles (Klute, 1986; Rawls et al., 1991; Wösten et al., 2001). Organic matter content can also affect the water retention function of a soil (Rawls et al., 2003), and in turn the hydraulic conductivity, which is a function of the soil water content (Klute and Dirksen, 1986). Both the newly developing A horizon soil of the surface legacy sediment horizon and the underlying, mid-layer legacy sediment horizon have silt loam textures (Table 2), with statistically similar particle size distributions, being composed of ~10-12% sand, ~73-76% silt, and ~12-15% clay. The buried relict A horizon soils have a textural classification of loam, with significantly higher sand (at ~28-38%) and lower silt (at ~46-50%) contents than the upper two legacy sediment horizons. The mid-layer legacy sediment horizon had greater mean bulk density than both the surface legacy sediment and relict A soil horizons, which corresponded to lower calculated porosity as compared to the other two horizons (Table 2).

Soil water retention curves relating experimentally measured soil volumetric water content and soil water potential for the three soil horizons at BSR (Fig. 5a) were similar for the surface and mid-layer legacy sediment horizons, with the mid-layer legacy sediment horizon having slightly higher soil volumetric water contents across the different soil water potentials. Though both the upper soil horizons had the same silt loam texture, the mid-layer legacy sediment horizon had a greater mean bulk density and lower porosity, as compared to the surface legacy sediment horizon. As such, the mid-layer legacy sediment horizon likely has a higher volume of smaller pores that can hold water more tightly, which equates to a higher soil volumetric

water content between -6 and -1500 kPa of tension. The relict A horizon soil showed the highest soil volumetric water content at the higher soil water potentials (i.e. less negative, in the range of -1 to -100 kPa), where soil structure predominately influences the shape of the soil water retention curve. Organic matter content, which can impact both soil structure and adsorption properties (Rawls et al., 2003), is greater in the deeper relict A horizon soils (Table 2), possibly explaining the initially high soil volumetric water contents at high soil water potentials. Below -100 kPa the relict A horizon soils hold less water than the surface and mid-layer legacy sediment horizons, likely due to the greater influence soil texture exerts at such lower soil water potentials (Dane and Hopmans, 2002a).

Soil hydraulic parameters were determined by fitting the soil water retention data to the van Genuchten soil hydraulic model (Table 3). Saturated hydraulic conductivity (Ks) values for the BSR soil horizons (Table 3) were obtained from Rawls et al. (1998) who used over 900 reported measurements to assemble Ks classification tables according to USDA soil texture classes and calculated porosity values. The soil hydraulic parameters and saturated hydraulic conductivity values were used to generate a plot of hydraulic conductivity versus soil water potential for the three soil horizons, representing both saturated and unsaturated flow (Fig. 5b). Soil water potentials at, or near zero, typically characterize the saturated flow region, while those lower than -10 kPa typically characterize the unsaturated flow region. The estimated saturated hydraulic conductivity (Table 3) (i.e. at a soil water potential of 0 kPa) was the same for both the surface and mid-layer legacy sediment horizons. However, as the soil water potential levels decrease (corresponding to lower moisture contents) the hydraulic conductivity of the mid-layer legacy sediment horizon drops below the values of the surface legacy sediment horizon. This, again, can be explained by the different mean bulk densities and porosities of the two upper soil horizons. The surface legacy sediment horizon likely contains more large pores that are water-filled when the soil water potential is high, but most of these will have been emptied by the time the soil water potential becomes very low, at about -100 kPa. Thus, at lower soil water potential values the hydraulic conductivity of the mid-layer legacy sediment horizon becomes greater than the surface legacy sediment horizon, probably due to the presence of a higher proportion of small pores that are still water-filled. The relict A horizon soil had the overall lowest hydraulic conductivities, both in the saturated and unsaturated flow regions. Higher organic matter content in the relict A horizon soil may explain its lower bulk density. Further, its low bulk density indicates that the horizon is likely a well-sorted soil, and though it has the highest percentage of sand, the low hydraulic conductivities at higher soil water potentials (higher moisture contents) suggest that the sand is more fine-grained than that found in the two upper soil horizons.

Along a legacy sediment strewn stream channel, like BSR, where the surface water and groundwater are disconnected, unsaturated flow will largely control water movement. The water table at BSR tends to fluctuate near the boundary between the buried relict A horizon soil and the basal gravels, with saturated conditions likely only occurring during high intensity precipitation events due to a rising water table or the infiltration of water into the surface legacy sediment horizon and its subsequent percolation through the soil profile. The mid-layer legacy sediment, which has the greatest hydraulic conductivity

at low soil water potentials, could potentially continue to contribute to unsaturated flow long after it has stopped in the surface legacy sediment and relict A soil horizons. This suggests that though it has the potential to retain the greatest amount of $^{15}NO_3^-$, the mid-layer legacy sediment horizon could leach more $^{15}NO_3^-$ over time that is either lost laterally to the stream, or via vertical transport into the buried relict A horizon soil that has low microbial activity (Weitzman et al., 2014), where there is little potential for immobilization or denitrification. However, though the three soil horizons each showed a range of hydraulic conductivities over varying soil water potentials, all were <1.5 cm $h^{-1}$ under saturated flow conditions, which corresponds to a flow rating of slow to moderately slow (Hazelton and Murphy, 2007). During unsaturated flow, which is the more typical flow regime at BSR, the hydraulic conductivities are categorized as very to extremely slow (Hazelton and Murphy, 2007). The water movement through the soils at BSR is of such a slow rate that it is unlikely that the high $NO_3^-$ leaching is caused by a hydrological bypass effect (i.e. preferential flow). This suggests that the relict A horizon soil core for which the release of $^{15}NO_3^-$ versus native $NO_3^-$ over time was analysed (mentioned above) likely experienced artificially-created preferential flow. The slow measured hydraulic conductivities further provide evidence that the low $NO_3^-$ retention ability of the BSR soils is due to the manifestation of N saturation conditions, as opposed to too short of an interaction time with soil particles (i.e. hydrological bypass).

## 4. Conclusions

Contrasting the dominant sources of variation in $NO_3^-$ retention capacity in the soils of BSR revealed 3 key results: (1) Geomorphic: Surface legacy sediment horizons do not retain excess $NO_3^-$ inputs well; (2) Climatic: Exposed relict A horizon soils experience the largest drought-induced $NO_3^-$ flush following sequential leaching; and 3) Cultural: Restoration that hydrologically reconnects the stream to its floodplain via legacy sediment removal may lead to an initial decrease in $NO_3^-$ retention capacity.

Low initial soil $^{15}NO_3^-$ retention (<17%) in all three soil horizons (surface legacy sediment, mid-layer legacy sediment, and relict A horizon soil) that was largely balanced by high $^{15}NO_3^-$ recovery in soil leachate material suggests that the soils of BSR are already $NO_3^-$ saturated, and, more specifically, are characterized by kinetic N saturation. The soil horizons are still active, as shown by their ability to retain some N inputs, but the large, simultaneous loss of N via $NO_3^-$ leaching suggests that the input rates of new N are exceeding the soils' total sink strength. Low hydraulic conductivity values indicate that rapid transport of $NO_3^-$ through the soil profile is unlikely. However, evidence of $NO_3^-$ flushing following the rewetting of dry soil, which was especially large in the mid-layer legacy sediment and relict A soil horizons, suggests that a fluctuating water table that causes saturation from the relict A horizon soil upward could potentially release stored $NO_3^-$ into the nearby waterway. Overland flow that occurs when surface soils are saturated, could also similarly result in a large flush of $NO_3^-$ being added to

the stream. There is less of a concern that $NO_3^-$ will be released when surface soils are rewet from above (i.e. following a rain event), because $NO_3^-$ storage in the surface legacy sediment is low, and slow percolation through the soil profile of surface losses of $NO_3^-$ could be counterbalanced by the higher retention capacity of the mid-layer legacy sediment horizon below.

Restoration that exposes the relict A horizon soil by removing the overlying legacy sediment seeks to increase groundwater-surface water interaction, which could potentially lead to higher $NO_3^-$ retention over the long-term. However, the short-term response to restoration efforts, as reflected in our data, may cause initially high $NO_3^-$ losses due to increased soil disturbance to the relict A horizon soil, as well as the removal of the mid-layer legacy sediment horizon, which showed a greater retention capacity for $NO_3^-$. The flow velocity of surface and groundwater could also potentially change as they adjust to the new surface level. It is likely the flow rates would increase, which in turn could promote further kinetic saturation conditions in the exposed relict A horizon soil, with the $NO_3^-$ input rates exceeding the $NO_3^-$ retention rates. It will be important to continue to monitor changes in $NO_3^-$ production and losses at BSR following restoration to determine if the newly exposed relict A horizon soils will ever become more biogeochemically active, like non-buried surface A horizon soils. Investigating other N loss pathways that may become more predominant in the restored wetland, like nitrous oxide fluxes, will also be necessary in order to fully understand the efficacy of restoration efforts based on the removal of legacy sediments.

## Author Contributions

J. N. Weitzman was the lead researcher on this study, performing data collection and analysis, drafting the manuscript, and helping to conceive of the experimental design and secure funding. J. P. Kaye helped with the conception of the experimental design and securing funding, critically revised the manuscript, and provided mentoring throughout the study.

## Competing Interests

The authors declare that they have no conflict of interest.

## Acknowledgements

We thank Terry Troutman, David Otto, and the field crew of the USDA-ARS Pasture Systems and Watershed Management Research Unit for their assistance with soil column construction and sample collection. This work was supported by the United State Geological Survey (USGS) 104B Grant administered by the Pennsylvania Water Resources Research Center (PA-WRRC) under Grant No. 2011PA156B. J. N. Weitzman also received support from the USDA-NIFA pre-doctoral fellowship program (2015-67011-22796).

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

## Tables

Table 1. Total soil carbon, total soil nitrogen, soil extractable ammonium ($NH_4^+$-N), soil extractable nitrate ($NO_3^-$-N), soil $NH_4^+$-N leached, soil $NO_3^-$-N leached, and $\delta^{15}N$ expressed as averages across soil depths (surface legacy sediment, mid-layer legacy sediment, and relict A horizon soil) and sampling time (field capacity vs. pre-drought vs. post-drought samples - $^{15}NO_3^-$ tracer added after field capacity measurements). These means (and one standard error in parentheses; n = 5 for each depth and sampling time) represent concentrations measured on fresh soils.

| | | Depth‡ | | |
|---|---|---|---|---|
| | Sampling Time† | Surface Legacy | Mid-Layer Legacy | Relict A Horizon |
| | | | $g\ m^{-2}$ | |
| Total Soil C | Pre-Drought | 3501 (70)$^{a}$ | 3373 (296)$^{a}$ | 6999 (461)$^{b}$ |
| | Post-Drought | 3768 (188)$^{a}$ | 3152 (50)$^{a}$ | 7796 (1177)$^{b}$ |
| | | | $g\ m^{-2}$ | |
| Total Soil N | Pre-Drought | 406 (6)$^{a}$ | 351 (34)$^{a}$ | 646 (44)$^{b}$ |
| | Post-Drought | 426 (15)$^{a}$ | 354 (7)$^{a}$ | 772 (108)$^{b}$ |
| | | | $g\ m^{-2}$ | |
| Soil Extractable | Pre-Drought | 0.51 (0.02)$^{ab}$ | 0.27 (0.01)$^{a}$ | 1.33 (0.39)$^{b}$ |
| $NH_4^+$-N | Post-Drought | 0.86 (0.35) | 0.50 (0.12) | 1.07 (0.23) |
| | | | $g\ m^{-2}$ | |
| Soil Extractable | Pre-Drought | 4.01 (0.39) | 3.59 (0.66) | 3.53 (0.28)$^{A}$ |
| $NO_3^-$-N | Post-Drought | 2.80 (0.56) | 2.24 (0.10) | 2.10 (0.19)$^{B}$ |
| | | | $g\ m^{-2}$ | |
| Soil | Field Capacity | 0.02 (0.01)$^{ab}$ | 0.00 (0.00)$^{a}$ | 0.03 (0.01)$^{b}$ |
| $NH_4^+$-N | Pre-Drought | 0.00 (0.00) | 0.00 (0.00) | 0.04 (0.01) |
| Leached | Post-Drought | 0.01 (0.00) | 0.01 (0.00) | 0.01 (0.00) |
| | | | $g\ m^{-2}$ | |
| Soil | Field Capacity | 6.80 (0.25)$^{a,A}$ | 3.15 (0.47)$^{b,A}$ | 1.30 (0.24)$^{c}$ |
| $NO_3^-$-N | Pre-Drought | 2.43 (0.65)$^{B}$ | 1.71 (0.17)$^{B}$ | 1.73 (0.10) |
| Leached | Post-Drought | 2.66 (0.58)$^{a,B}$ | 0.71 (0.16)$^{b,B}$ | 1.20 (0.10)$^{b}$ |

| | | | | |
|---|---|---|---|---|
| | | | L | |
| Total Volume | Pre-Drought | 10.32 (0.11)$^{A}$ | 10.50 (0.12)$^{A}$ | 10.00 (0.35) |
| Leached | Post-Drought | 8.16 (0.68)$^{B}$ | 7.46 (0.66)$^{B}$ | 8.70 (0.85) |
| | | | ‰ | |
| | Initial§ | +7.11 (0.20)$^{a,A}$ | +5.66 (0.38)$^{a,A}$ | +3.65 (0.52)$^{b,A}$ |
| $\delta^{15}N$ | Pre-Drought | +53.15 (5.98)$^{a,B}$ | +159.91 (23.16)$^{b,B}$ | +72.43 (10.86)$^{a,B}$ |
| | Post-Drought | +20.29 (2.63)$^{a,A}$ | +33.32 (3.57)$^{b,A}$ | +6.64 (1.18)$^{c,A}$ |

†For a given sampling time, values with different superscript lowercase letters represent statistically significant ($P < 0.05$) differences with depth.

‡For a given depth, values with different superscript uppercase letters represent statistically significant ($P < 0.05$) differences with sampling time.

§Initial $\delta^{15}N$ values are based on soil samples collected from stream banks in September 2010 which were discussed in Weitzman et al., 2014.

Table 2. Bulk density, porosity, texture, and particle size distribution classes (sand, silt, and clay) expressed as averages across soil depths (surface legacy sediment, mid-layer legacy sediment, and relict A horizon soil). Values are means (n=8 for bulk density and porosity, and n=5 for organic matter content and particle size classes) and one standard error in parentheses.

| Depth† | Bulk Density‡ (g $cm^{-3}$) | Porosity§ | Organic Matter‡ (%) | Sand (%) | Silt (%) | Clay (%) | Texture |
|---|---|---|---|---|---|---|---|
| Surface Legacy | 0.78 (0.02)[a] | 0.70 (0.01)[a] | 4.25 (0.30)[a] | 12 (0.4)[a] | 73 (0.9)[a] | 14 (0.6) | Silt Loam |
| Mid-Layer Legacy | 1.06 (0.05)[b] | 0.59 (0.02)[b] | 3.49 (0.21)[b] | 11 (0.5)[a] | 76 (1.2)[a] | 13 (0.9) | Silt Loam |
| Relict A Horizon | 0.76 (0.09)[a] | 0.71 (0.04)[a] | 4.84 (1.77)[ab] | 33 (5)[b] | 48 (1.9)[b] | 18 (4.9) | Loam |

†For each given property, values with different superscript lowercase letters represent statistically significant ($P < 0.05$) differences among soil horizons.

‡Bulk density and organic matter content values reproduced from Weitzman et al. (2014).

§All three soil horizons had a particle density of 2.6 g $cm^{-3}$ as determined via a gas pynctnometer by Merritts et al., 2010.

Table 3. Soil hydraulic parameters† derived from soil water retention curves fitted to the van Genuchten soil hydraulic model‡ using the SWRC Fit program§ for each soil depth (surface legacy sediment, mid-layer legacy sediment, and relict A horizon soil).

| Depth | $K_s$¶ (cm $h^{-1}$) | $\theta_s$ ($cm^3$ $cm^{-3}$) | $\theta_r$ ($cm^3$ $cm^{-3}$) | $\alpha$ ($cm^{-1}$) | *m* | *n* |
|---|---|---|---|---|---|---|
| Surface Legacy | 1.44 | 0.34 | 0.19 | 0.17 | 0.31 | 1.46 |
| Mid-Layer Legacy | 1.44 | 0.37 | 0.03 | 0.17 | 0.11 | 1.13 |
| Relict A Horizon | 0.39 | 0.44 | 0.12 | 0.17 | 0.29 | 1.41 |

†Ks, saturated hydraulic conductivity; $\theta_s$, saturated soil water content; $\theta_r$, residual soil water content; $\alpha$, *m*, and *n*, curve-fitting parameters.

‡van Genuchten, 1980

§Seki, 2007

¶Saturated hydraulic conductivity obtained from Rawls et al., 1998 (from classification by USDA soil texture classes and porosity).

## Figures

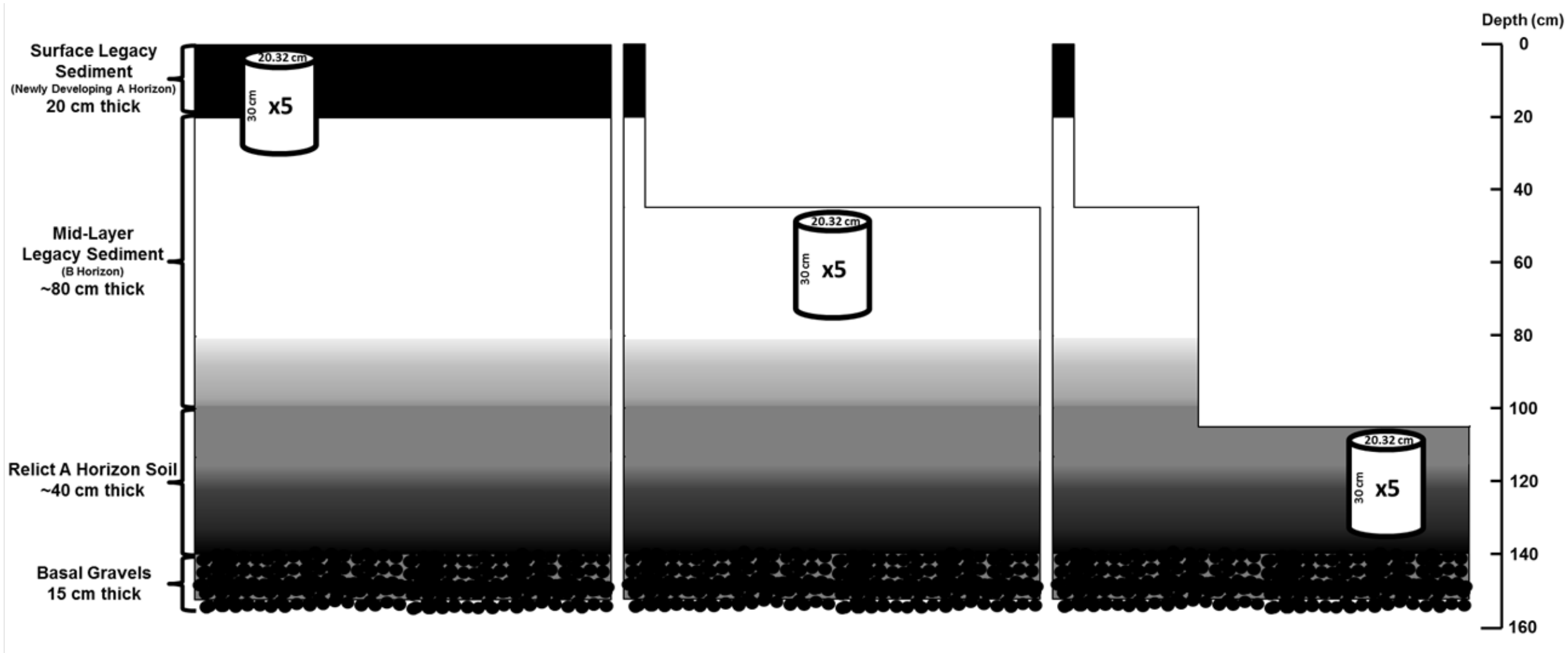

Figure 1. Step-wise sampling scheme used for soil core collection at Big Spring Run. Five intact soil cores (20.32 cm diameter and 30 cm length) were collected for each of the three soil horizons of interest (surface legacy sediment, mid-layer legacy sediment, and relict A horizon soil). Surface legacy sediment cores were collected in one area, then the surface legacy sediment was removed from an adjacent area to sample the underlying legacy sediment, and then finally the surface and mid-layer legacy sediment were removed from a third adjacent area to sample the underlying relict A horizon soil.

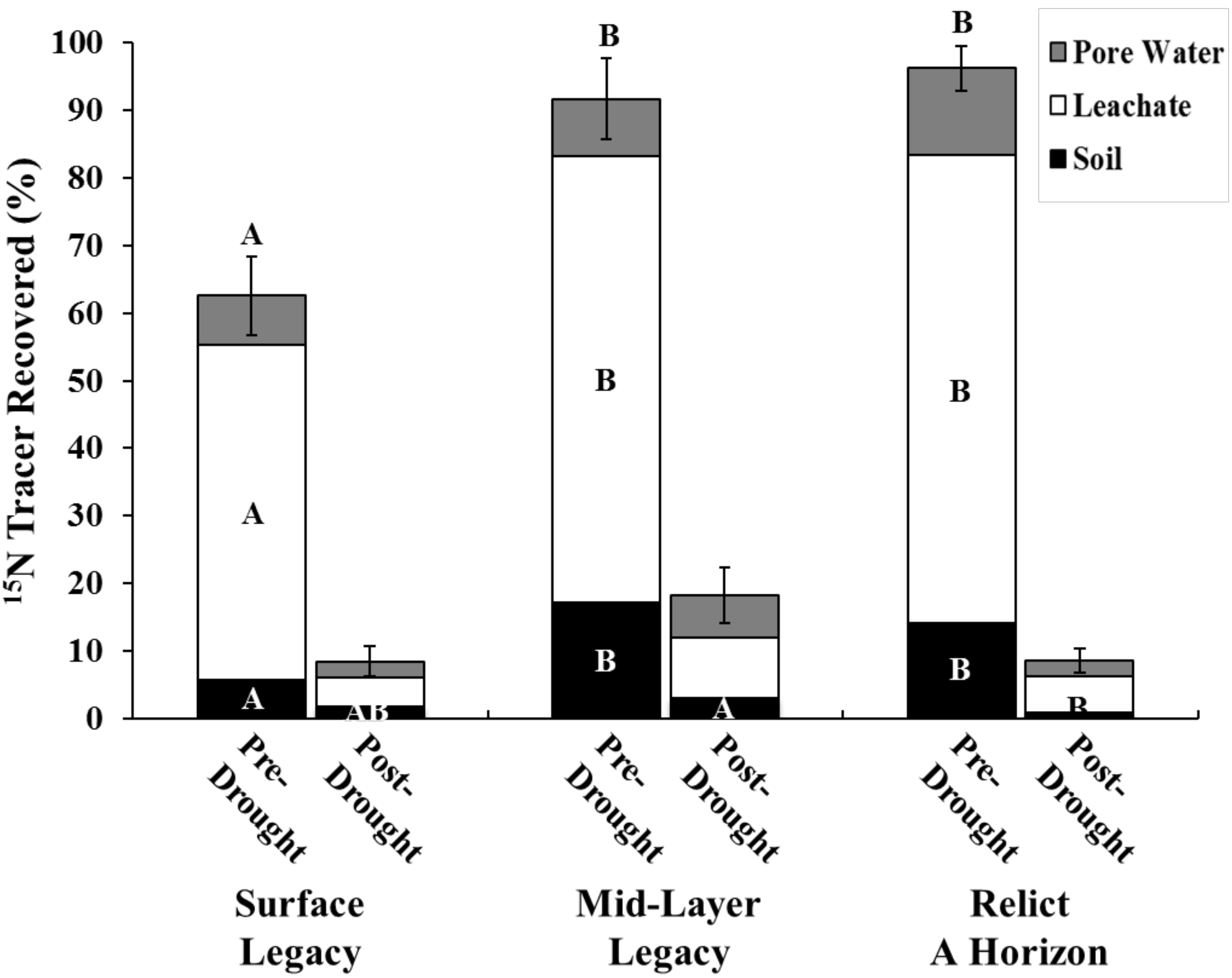

Figure 2. Amount of tracer $^{15}N$ (as $^{15}NO_3^-$) (%) recovered in the soil (black bars), leachate (white bars), and pore water (gray bars) pools pre-drought and post-drought. Vertical bars denote one standard error of the mean for the total of the three pools combined (n = 5). For a given sampling time (i.e. pre-drought or post-drought) different letters above the standard error bars represent statistically significant ($P < 0.05$) differences in the total recovery of $^{15}N$ (i.e. soil + leachate + pore water) across soil horizons. For a given $^{15}N$ pool (i.e. soil or leachate or pore water) and sampling time (i.e. pre-drought or post-drought) different letters represent statistically significant ($P < 0.05$) differences across soil horizons.

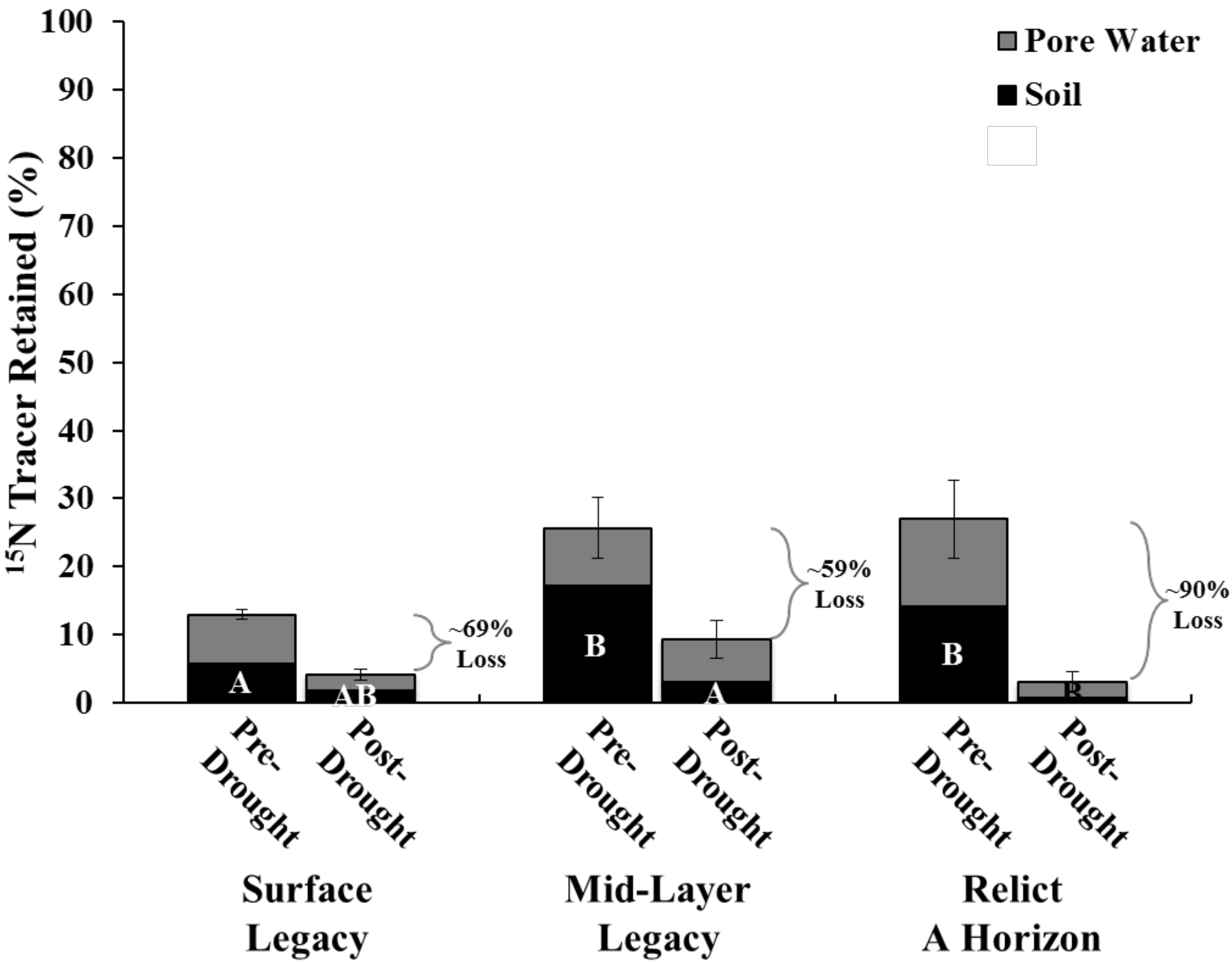

Figure 3. Amount of tracer $^{15}N$ (as $^{15}NO_3^-$) (%) retained in the soil (black bars) and pore water (gray bars) pools pre-drought and post-drought. Vertical bars denote one standard error of the mean for the two pools combined (n = 5). Retention of $^{15}N$ was not found to be statistically significant among soil horizons when the two pools were combined. But, for a given pool and sampling time different letters represent statistically significant (*P* < 0.05) differences across soil horizons. The average pulsed loss of $^{15}N$ (%) from each soil horizon signifies the percent decrease in $^{15}N$ retention (soil + pore water pools combined) between the two moisture conditions (pre-drought and post-drought).

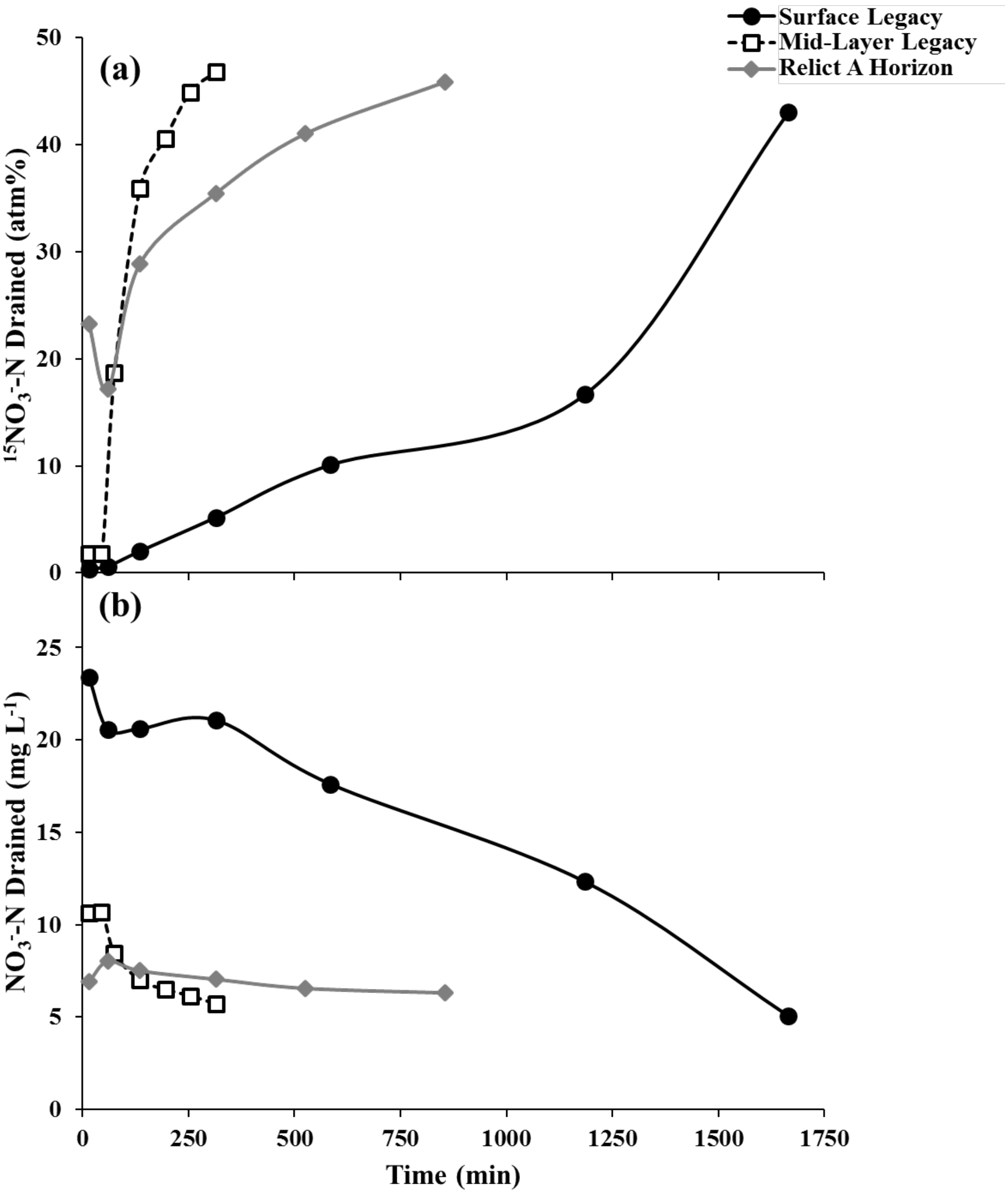

Figure 4. Time series release curves depicting the release of leachate (a) $^{15}NO_3^-$ and (b) $NO_3^-$ over time from one soil column of each of the soil horizons of interest (surface legacy sediment, mid-layer legacy sediment, and relict A horizon soil) during pre-drought conditions.

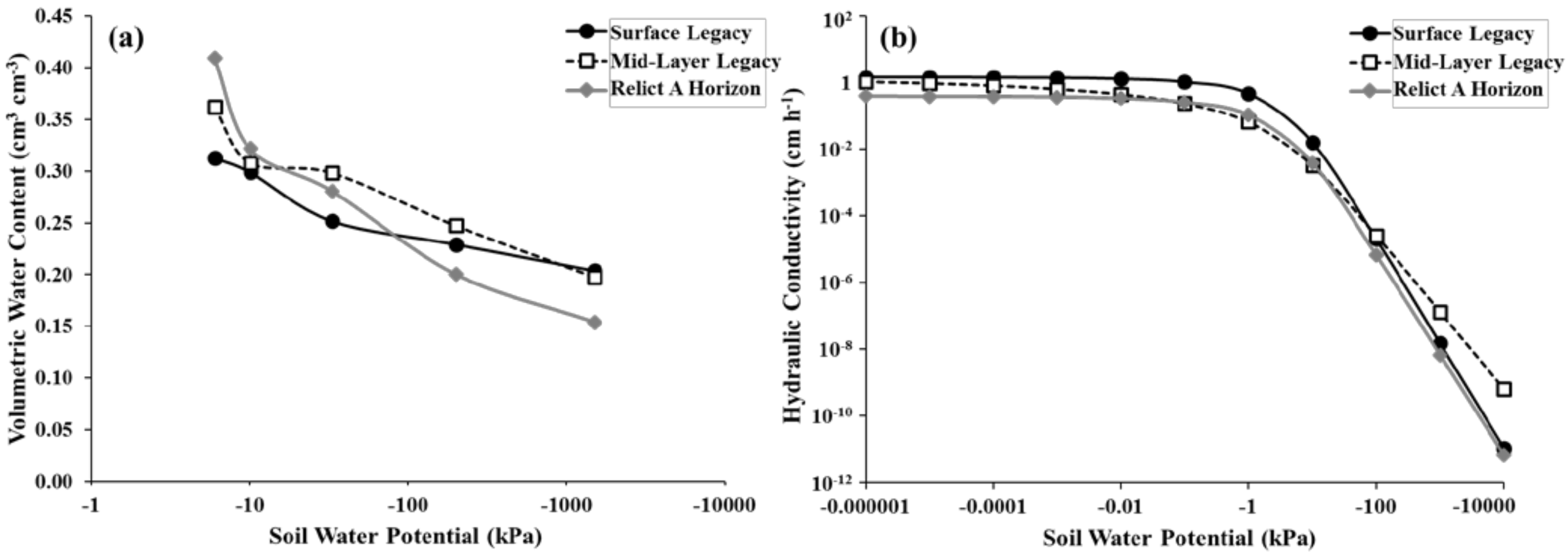

Figure 5. (a) Soil water retention curves relating experimentally measured soil volumetric water content and soil water potential and (b) hydraulic conductivity curves derived from the van Genuchten soil hydraulic model for the three soil horizons of interest (surface legacy sediment, mid-layer legacy sediment, and relict A horizon soil) at Big Spring Run.