# Peer review of "Nitrate retention capacity of milldam-impacted legacy sediments and relict A horizon soils"

_SOIL, 2016_

## Referee Comment (RC1) · Anonymous Referee #1 · 18 Oct 2016

General comments

The topic of the manuscript is suitable for the journal Soil. It reports on nitrate leaching from two soil horizons originating from sediments behind a milldam and from an older horizon buried under them. From the regional prevalence and from the high leaching rates observed, this study is certainly justified. The topic is very well explained in the introduction, which describes not only the specific questions of the study but also the geographic and historic context. The presentation of the site is extensive and very well written.

The main value of the experiment lies in the fact that such soil profiles have barely been studied in this manner. This value is, however, limited in the sense that it covers

only one site. It is therefore not obvious to draw conclusions for other sites impacted by milldam sediments.

The study was conducted on soil columns, which is an efficient way to measure leaching. Nitrate added to these columns was labelled with 15N, which allows to distinguish it from soil-derived nitrate. It would have been useful to add also a biologically inert tracer like bromide, to be able to distinguish between physical and biological processes. This would have especially been useful to ensure a better interpretation about preferential water flow, which may have been favoured by the mechanical extraction of the soil columns, as the authors note. A dye may also have been useful to answer this question. For the interpretation of the flow regime, the authors analysed the texture of the soil and estimated hydraulic conductivities. However, they do not describe the structure of the soil, nor if there were roots in the soil columns. In spite of the weight of the sediments, the buried, relict A horizon was found to have a high porosity, but it had a low hydraulic conductivity: this may be due to an horizontal structure within this horizon, with some less permeable layer. It is at least not likely that a former A horizon would be homogeneous over 30 cm in depth (even if it had been ploughed, then at that time certainly not so deep). Without a proper description of the structure, interpretations about the flow regime are difficult. It would certainly be useful to compare the measured water infiltration rates with those predicted by the parameters of table 3.

The results about general soil properties are given after the results of nitrate leaching. The contrary would certainly facilitate the presentation and discussion.

Further, the question of N saturation (especially its kinetic aspect) would require a comparison of the applied amounts (approximately 0.5 g/m2 if my calculation is correct) with nitrification rates and with nitrate in atmospheric deposition. These rates are unfortunately not quantified here. The discussion about assimilation and remineralisation is essentially justified, but quantitatively it is likely to play only a minor role: it is less likely that N is assimilated and remineralised and nitrified and leached than just leached. The relative importance of direct leaching versus leaching after mineralisation
and nitrification should be discussed in the light of the tracer fraction in the leachate, i.e. the molar ratio of tracer N to total N in leached nitrate.

Finally, the discussion about a co-saturation of C and N is not convincing in this case. At the basis of this concept, there is the work of Cleveland & Liptzin (Biogeochemistry 85 (2007): 235-252), who give for such a co-saturation a C/N ratio of the soil near 14. Here, however, C/N ratios are around 9 or 10 (calculated from table 1). This means that the amount of C present in the soil would still give much "room" for N immobilisation. Further, a co-saturation is likely to be limited to soils which are well drained, have a sufficient pH and are warm enough. While the third condition is certainly fulfilled here, the other ones would need to be discussed: what is the pH in these soil horizons, and are there signs of anaerobicity that could hinder mineralisation?

Details

Introduction

Page 3, line 27: the abbreviation BSR should be defined here in the introduction (it is defined only later).

P. 3, L. 27: "created" is perhaps not the best wording for nitrate. And there is also nitrate which is not formed in the soil but brought by atmospheric deposition.

Material and methods

P. 6, L. 14: was the soil cultivated? If there were plants, how were they removed? Was there a litter layer, and if yes was it removed?

P. 7, L. 12: the amount of water corresponds to 92 mm, which is high but not impossible for a single precipitation event.

P. 7, L. 28: "g soil-1" should be written with a parenthesis: (g soil)-1, otherwise it would be like only "soil" and not "g" is at the power -1.

P. 8, L. 16: atom% enrichment is not defined. This would be necessary to understand

correctly the given equations.

P. 9, L. 8: multiplying by 100 is for a unit transformation (from a simple ratio to percent): such unit transformations are not an essential part of an equation and should be given only if it explicitly presented a such.

P. 9, L. 22: it would be useful to write in a couple of words what is the working principle of this software.

Results and discussion

P. 11, L. 23: "vice versa"

P. 11, L. 29: better name a dimension by its proper name than indirectly by it units, i.e. better "abundance" than "atom%".

P. 12, L. 26: "too slow" is relative to the speed of leaching, i.e. the interpretation could as well be that leaching is too fast, especially in the case of preferential flow.

P. 13, L. 7: adsorption of ammonium is certainly also a reason why the concentration in solution are lower.

P. 14, L. 3: this repeats what has bee written above.

P. 15, L. 2: quite a long sentence.

Tables and figures

Tab. 1: the legend of the figure mentions "concentrations" but the data are amounts per area, and it is not clear if these amounts are only for the 30 cm columns or if they are extrapolated for the whole corresponding horizon in the field.

Tab. 2: it is astonishing that the relict A horizon has as much organic matter than the other horizons while it has (according to table 1) a much higher C content (but this may be related to the question above).

Fig. 2: all the information in this figure is already contained in figure 1. It should thus

be deleted.

Fig. 4: what are "atm%" in the legend of the Y axis? Are these atom%? The graph would be easier to read if it would give a tracer fraction (as defined above).

Conclusion

The manuscript discussed here starts very well and lets the reader hope for an exciting story. However, it shows then some limitations in the material and methods and ends with a discussion that misses some important aspects.

---

## Referee Comment (RC2) · Anonymous Referee #2 · 25 Jan 2017

This article is suitable for publication in soil. It describes the difference in NO3 movement between different horizons of soil which were either present prior to the milldam being created, or deposited as sediments.

The manuscript clearly describes the context of the work, and provides a good summary into the history and formation of these anthropogenic soil profiles. The site is well described, and methodological approaches used are appropriate.

While the results and discussion section clearly outlines the results obtained, it is somewhat light on regarding discussion. I would encourage the authors to provide more insights into the processes driving the results obtained, with support from appropriate literature; furthermore, there is only a limited connection of this research with the

literature, based on the limited citations present in the discussion.

The presentation of the soil properties in the results section is quite comprehensive – maybe a bit too much so. I would like the authors to consider abridging this section and focussing on the results immediately relevant to NO3 transfer. Moreover, if this section were to presented prior to the NO3 leaching results, it would provide a greater context for discussion of those results, and the opportunity to finish off the discussion with a clear description of how the results link together, what you have learnt about each horizon, and how does all this information fit together to understand how NO3 would flow through this system if it was undisturbed?

Following on, some more discussion about restoration would have been helpful for those not in that space – what are the environmental benefits of restoration – magnitude changes in NO3 losses?

Finally, it was somewhat confusing to get through most of the paper, only to read that one whole section of results (time series 15NO3 vs native NO3) is likely to have been compromised due to preferential flow. I would encourage the authors to consider the value of retaining the time series experiment in the manuscript – how much value does it actually add, or, would it be missed if it wasn't there?

Detail:

Intro: P2 L7: The "Williams 2000" citation was not found in the list of references P4 L5: I'm not sure the cultural question regarding restoration was actually addressed in this manuscript. If this is an important aspect, a section at the end demonstrating the predicted in-situ effects of restoration would be of value. P5 L8: The "Brush 2008" citation was not found in the list of references.

Materials and Methods: P7 L6: Why was K2SO4 used as N-free water? P7 L6: How was the pore volume estimated? P7 L27: what was the soil:solution ratio of the 2M KCl extraction, and what were the conditions for mixing? P9:L1: The 15N recovery

vs retention section is not well explained. Please articulate more clearly the value of presenting the results in both ways, or consolidate Fig 2 and 3, which appear to show equivalent results.

Results and discussion: P11 L23: please spell out "atm%" P13 L9: If it is proposed that low NO3 uptake is the reason for the large NO3 leaching losses, please discuss some of the processes which may be governing NO3 uptake, and why these are low in this soil - . . . how does this compare with other milldams or equivalent textured soils? P14 L24: I'm not familiar with the term "well-sorted soil". Conclusions: P15 L20: The comment regarding restoration of the site may lead to an initial decrease in NO3 retention capacity – some comments around the magnitude and importance of this proposed decrease would be of value – how does it rate compared to the landscape as a whole?

References: - Castellano and Kaye (2009) not mentioned in paper - Merrits et al (2010) not mentioned in paper.

---

## Author Comment (AC1) · 23 Feb 2017

Anonymous Referee #1: Received and published 18 October 2016

*Author responses are italicized in blue below the reviewer comments.*

General comments:

The topic of the manuscript is suitable for the journal Soil. It reports on nitrate leaching from two soil horizons originating from sediments behind a milldam and from an older horizon buried under them. From the regional prevalence and from the high leaching rates observed, this study is certainly justified. The topic is very well explained in the introduction, which describes not only the specific questions of the study but also the geographic and historic context. The presentation of the site is extensive and very well written.

The main value of the experiment lies in the fact that such soil profiles have barely been studied in this manner. This value is, however, limited in the sense that it covers only one site. It is therefore not obvious to draw conclusions for other sites impacted by milldam sediments.

*Response: We agree that the findings at BSR (just one site impacted by milldam sediments) may not be representative of all other sites similarly impacted by milldam sediments. Therefore, we have added a caveat in the results and discussion section bringing up this point, and include the comment that further research at multiple sites impacted by milldam sediments is required to better understand if the low $NO_3^-$ retention we found in legacy sediments is specific to the BSR site or whether it is characteristic of all sites similarly impacted by legacy sediments.*

The study was conducted on soil columns, which is an efficient way to measure leaching. Nitrate added to these columns was labeled with 15N, which allows to distinguish it from soil-derived nitrate. It would have been useful to add also a biologically inert tracer like bromide, to be able to distinguish between physical and biological processes. This would have especially been useful to ensure a better interpretation about preferential water flow, which may have been favoured by the mechanical extraction of the soil columns, as the authors note. A dye may also have been useful to answer this question.

*Response: We agree that adding a biologically inert tracer to distinguish between physical and biological processes, or a dye to better explain the main flow regimes through the soil columns, would have been useful techniques to employ during this study. Unfortunately, we did not utilize these methods during our study, but we did add a comment in the conclusions section that data interpretation could be strengthened in the future if a co-tracer or dye is added when conducting similar soil column leaching/retention studies.*

For the interpretation of the flow regime, the authors analysed the texture of the soil and estimated hydraulic conductivities. However, they do not describe the structure of the soil, nor if there were roots in the soil columns. In spite of the weight of the sediments, the buried, relict A horizon was found to have a high porosity, but it had a low hydraulic conductivity: this may be due to an horizontal structure within this horizon, with some less permeable layer. It is at least not likely that a former A horizon would be homogeneous over 30 cm in depth (even if it had been ploughed, then at that time certainly not so deep). Without a proper description of the

structure, interpretations about the flow regime are difficult. It would certainly be useful to compare the measured water infiltration rates with those predicted by the parameters of table 3.

*Response: The reviewer makes a good point that soil structure and the presence/absence of roots in the different soil horizons are important parameters that might impact our interpretations of different flow regimes. We added a description of both these parameters to the Soil Properties section of the text and drew on them to enhance our interpretations. However, we do not have measured water infiltration rates, which is why we needed to use the predicted parameters of Table 3.*

The results about general soil properties are given after the results of nitrate leaching. The contrary would certainly facilitate the presentation and discussion.

*Response: We rearranged these two sections. Now we discuss the general soil properties prior to presenting the results of the nitrate leaching experiment.*

Further, the question of N saturation (especially its kinetic aspect) would require a comparison of the applied amounts (approximately 0.5 g/m2 if my calculation is correct) with nitrification rates and with nitrate in atmospheric deposition. These rates are unfortunately not quantified here. The discussion about assimilation and remineralisation is essentially justified, but quantitatively it is likely to play only a minor role: it is less likely that N is assimilated and remineralised and nitrified and leached than just leached. The relative importance of direct leaching versus leaching after mineralization and nitrification should be discussed in the light of the tracer fraction in the leachate, i.e. the molar ratio of tracer N to total N in leached nitrate.

*Response: We agree that to definitively determine the extent of N saturation in the system would require a much larger dataset, with a more complete inventory of inputs and outputs, which is beyond the scope of the current study. We focused on learning more about $NO_3^-$ retention within the different soil horizons of BSR, and believe that our findings can best be interpreted using N saturation theory. As a first assessment we believe our hypothesis is reasonable, and acknowledge that more work is needed to quantitatively assert whether the low $NO_3^-$ retention can be attributed solely to the system's N saturation status. We added this line of thinking to the text. We also agree that direct leaching is likely much more important than leaching after mineralization and nitrification (as we discussed in the text), but followed the reviewer's comments and added some quantitative data to back up this by looking at the tracer fraction in the leachate.*

Finally, the discussion about a co-saturation of C and N is not convincing in this case. At the basis of this concept, there is the work of Cleveland & Liptzin (Biogeochemistry 85 (2007): 235-252), who give for such a co-saturation a C/N ratio of the soil near 14. Here, however, C/N ratios are around 9 or 10 (calculated from table 1). This means that the amount of C present in the soil would still give much "room" for N immobilisation. Further, a co-saturation is likely to be limited to soils which are well drained, have a sufficient pH and are warm enough. While the third condition is certainly fulfilled here, the other ones would need to be discussed: what is the pH in these soil horizons, and are there signs of anaerobicity that could hinder mineralisation?

*Response: Prior findings at BSR showed that the relict A horizon appears to be inefficient at utilizing new C inputs (Weitzman et al., 2014). Thus, while the C:N ratio of the relict A*

*horizon may be near 10, instead of near 14, as Cleveland and Liptzin (2007) give as the critical ratio at which co-saturation is likely to occur, the finding that the soil horizon cannot utilize new C inputs suggests that it acts similarly to a C-saturated soil. For this reason we believe that our comment about co-saturation of C and N holds. We did, however, add a similar explanation to the text to address the above comment.*

Details:

Introduction
Page 3, line 27: the abbreviation BSR should be defined here in the introduction (it is defined only later).

*Response: We added the full definition for the BSR abbreviation as "Big Spring Run (BSR) in Lancaster, Pennsylvania" at this first introduction.*

P. 3, L. 27: "created" is perhaps not the best wording for nitrate. And there is also nitrate which is not formed in the soil but brought by atmospheric deposition.

*Response: We changed the wording from "created" to "produced in situ" to refer specifically to the $NO_3^-$ that forms in the soil, as opposed to that which can accumulate via atmospheric deposition.*

Material and methods
P. 6, L. 14: was the soil cultivated? If there were plants, how were they removed? Was there a litter layer, and if yes was it removed?

*Response: We sampled the soil right next to Big Spring Run (<5 m from the edge of the high-cut stream bank; what is traditionally the riparian zone). The soil in this area was not cultivated and was characterized by tall grasses that were clipped close to ground-level (to a height of ~2.5 cm above the soil surface) to facilitate easier soil column extraction. Clipped grass and its associated roots remained in the surface soil columns for the leaching experiments. We added these details to the text.*

P. 7, L. 12: the amount of water corresponds to 92 mm, which is high but not impossible for a single precipitation event.

*Response: The amount of water corresponds to a rain event of ~370 mm (diameter of the cores = 20.32 cm, not the radius), which is more typical of an "extreme" rain event, as opposed to a normal precipitation event. Such an "extreme" rain event is more aligned with current climate change models that predict higher intensity rainfall events for the region, which could be followed by drier conditions.*

P. 7, L. 28: "g soil-1" should be written with a parenthesis: (g soil)-1, otherwise it would be like only "soil" and not "g" is at the power -1.

*Response: Parentheses were added around (g soil) to denote the correct unit raised to the -1 power.*

P. 8, L. 16: atom% enrichment is not defined. This would be necessary to understand correctly the given equations.

*Response: We realize that atom % enrichment can sometimes be confused with atom % excess, and could erroneously be interpreted to mean the enrichment above natural abundance. To avoid this confusion we chose to remove the word "enrichment" and instead use atom percent (shortened to atom %) throughout the paper. Here we define atom percent (atom %) as the absolute number of atoms of a given isotope (here $^{15}N$) in 100 atoms of the total element, i.e.*

*atom % $^{15}N$ = $\left(\frac{^{15}N}{^{14}N + ^{15}N}\right) \times (100)$. We included this definition to allow for more clarity.*

P. 9, L. 8: multiplying by 100 is for a unit transformation (from a simple ratio to percent): such unit transformations are not an essential part of an equation and should be given only if it explicitly presented as such.

*Response: We removed the "x 100" from the equation.*

P. 9, L. 22: it would be useful to write in a couple of words what is the working principle of this software.

*Response: The SWRC Fit program (Seki, 2007) performs non-linear fitting of soil water retention curves to a number of soil hydraulic models. The program automatically determines all the necessary initial conditions for the non-linear fitting, which allows users to simply input soil water retention curve data to obtain estimated hydraulic parameters. We added this explanation to clarify the working principle of the software, as suggested.*

Results and discussion
P. 11, L. 23: "vice versa"

*Response: We changed "vice-a-versa" to the wording "vice versa" as suggested.*

P. 11, L. 29: better name a dimension by its proper name than indirectly by it units, i.e. better "abundance" than "atom%".

*Response: We do not agree that "abundance" is the right term to use, but instead chose to keep atom %, which we defined earlier.*

P. 12, L. 26: "too slow" is relative to the speed of leaching, i.e. the interpretation could as well be that leaching is too fast, especially in the case of preferential flow.

*Response: This is a fair point. We clarified that the biological sinks may be "too slow" relative to the speed of leaching.*

P. 13, L. 7: adsorption of ammonium is certainly also a reason why the concentration in solution are lower.

*Response: We added that adsorption of ammonium to clay particles in the soil could also potentially explain why soil extractable concentrations of $NH_4^+$ were much lower than those of $NO_3^-$.*

P. 14, L. 3: this repeats what has been written above.

*Response: We removed the repeated explanation that organic matter content can impact the shape of the soil water retention curve.*

P. 15, L. 2: quite a long sentence.

*Response: We separated the one sentence into two sentences to reduce its original length and wordiness.*

Tables and figures
Tab. 1: the legend of the figure mentions "concentrations" but the data are amounts per area, and it is not clear if these amounts are only for the 30 cm columns or if they are extrapolated for the whole corresponding horizon in the field.

*Response: We replaced "concentrations" with the more accurate "mass per unit area" and clarified that the amounts in the table represent averages of the 30 cm soil core segments.*

Tab. 2: it is astonishing that the relict A horizon has as much organic matter than the other horizons while it has (according to table 1) a much higher C content (but this may be related to the question above).

*Response: Organic matter contents were reproduced from Weitzman et al., 2014. These correspond to different samples taken in a similar location. They may not correspond exactly to the soil columns, but the general trend should be similar.*

Fig. 2: all the information in this figure is already contained in figure 1. It should thus be deleted.

*Response: We believe both Figure 2 and Figure 3 are needed to understand the difference between 15N recovery vs. 15N retention. See longer explanation in response to Reviewer #2.*

Fig. 4: what are "atm%" in the legend of the Y axis? Are these atom%? The graph would be easier to read if it would give a tracer fraction (as defined above).

*Response: Atm% is atom percent. They should be interpreted in relation to the added 15N tracer which had a value of 60 atm%. We spelled out atom % in the figure and added the explanation about the tracer value to the legend for better clarification.*

Conclusion

The manuscript discussed here starts very well and lets the reader hope for an exciting story. However, it shows then some limitations in the material and methods and ends with a discussion that misses some important aspects.

*Response: As the first study to address the impact of legacy sediment effects on nitrate retention through the soil profile we are aware that there were some method limitations, and that more investigation is still needed. We have added these caveats to the conclusions, as discussed above, and have expanded the discussion to include more predictions about the long-term effects of restoration and other potential processes that may be limiting $NO_3^-$ retention in the legacy sediment impacted soils.*

---

## Author Comment (AC2) · 23 Feb 2017

Anonymous Referee #2: Received and published 25 January 2017

*Author responses are italicized in blue below the reviewer comments.*

General Comments:

This article is suitable for publication in soil. It describes the difference in NO3 movement between different horizons of soil which were either present prior to the milldam being created, or deposited as sediments.

The manuscript clearly describes the context of the work, and provides a good summary into the history and formation of these anthropogenic soil profiles. The site is well described, and methodological approaches used are appropriate.

While the results and discussion section clearly outlines the results obtained, it is somewhat light on regarding discussion. I would encourage the authors to provide more insights into the processes driving the results obtained, with support from appropriate literature; furthermore, there is only a limited connection of this research with the literature, based on the limited citations present in the discussion.

*Response: We expanded the discussion to better explain the potential processes driving our results by drawing more on associated literature, and included these citations throughout the text of the discussion section.*

The presentation of the soil properties in the results section is quite comprehensive – maybe a bit too much so. I would like the authors to consider abridging this section and focusing on the results immediately relevant to NO3 transfer. Moreover, if this section were to presented prior to the NO3 leaching results, it would provide a greater context for discussion of those results, and the opportunity to finish off the discussion with a clear description of how the results link together, what you have learnt about each horizon, and how does all this information fit together to understand how NO3 would flow through this system if it was undisturbed?

*Response: We believe the information in the soil properties section is necessary to understand $NO_3^-$ transfer within the different soil horizons, but abridged some details when possible. We also rearranged the presentation of results at the suggestion of both reviewers, with the soil properties section now appearing prior to the $NO_3^-$ leaching section.*

Following on, some more discussion about restoration would have been helpful for those not in that space – what are the environmental benefits of restoration – magnitude changes in NO3 losses?

*Response: We added more discussion about the potential environmental benefits of restoration, specifically related to expected changes in $NO_3^-$ levels over the long-term.*

Finally, it was somewhat confusing to get through most of the paper, only to read that one whole section of results (time series 15NO3 vs native NO3) is likely to have been compromised due to preferential flow. I would encourage the authors to consider the value of retaining the time series

experiment in the manuscript – how much value does it actually add, or, would it be missed if it wasn't there?

*Response: We suggest that artificially-created preferential flow may have impacted the results obtained in the relict A horizon soil, but still feel the results obtained for the other two soil horizons is likely representative of actual flow regimes. For this reason we believe including the $^{15}NO_3^-$ vs. native $NO_3^-$ time series data is still useful for interpretation purposes.*

Detail:

Intro:
 P2 L7: The "Williams 2000" citation was not found in the list of references

*Response: The "Williams, 2000" citation was added to the list of references.*

P4 L5: I'm not sure the cultural question regarding restoration was actually addressed in this manuscript. If this is an important aspect, a section at the end demonstrating the predicted in-situ effects of restoration would be of value.

*Response: We believe that we addressed the cultural question regarding restoration, in that restoration efforts that seek to remove the overlying legacy sediment, leaving only the once buried relict A horizon soil, could lead to an initial decrease in $NO_3^-$ retention capacity. We originally did not want to discuss predictions about the long-term effects of restoration, as it is outside the findings of our study, but as both reviewers commented that more discussion on the topic is needed we added text to both the discussion and conclusions sections expanding our predictions about the in-situ effects of restoration on $NO_3^-$ cycling over longer timescales.*

P5 L8: The "Brush 2008" citation was not found in the list of references.

*Response: The "Brush, 2008" citation had the wrong year attributed, it was actually "Brush, 2009", which can be found in the list of references. The citation in the text was corrected to "Brush, 2009".*

Materials and Methods:
P7 L6: Why was K2SO4 used as N-free water?

*Response: We used a very dilute (0.001 M) solution of $K_2SO_4$ as N-free water so as to add some electrolytes to the solution to better mimic additions expected in rainwater.*

P7 L6: How was the pore volume estimated?

*Response: Pore volume in the soil cores was estimated by multiplying the approximated average dry soil volume in the cores by the approximated average percent pore space. The average percent pore space was estimated as: % pore space = 100 - (100\*(bulk density/particle density)), assuming a particle density of 2.65 g $cm^{-3}$ and an average bulk density across the soil horizons of 1.00 g $cm^{-3}$. We believe this is too much detail to add to the text, but did add that pore space was estimated based on bulk density measurements.*

P7 L27: what was the soil:solution ratio of the 2M KCl extraction, and what were the conditions for mixing?

*Response: The 2M KCl extraction followed standard procedures (c.f. Bremner and Keeney, 1966) and was based on a 1:10 soil:extractant ratio – i.e. ~12-15 g of fresh weight soil (expected to be equivalent to ~10 g of dry weight soil) was measured into 100 mL of 2M KCl. Samples were extracted on a reciprocating horizontal mechanical shaker at room temperature for 1 hr, after which they were filtered through Whatman Grade 1 qualitative filter paper. Given this is a standard procedure we do not believe all the details are needed in the text. We did, however, add the soil:extractant ratio for reference.*

P9:L1: The 15N recovery vs retention section is not well explained. Please articulate more clearly the value of presenting the results in both ways, or consolidate Fig 2 and 3, which appear to show equivalent results.

*Response: Recovery of 15N shows how much of the added tracer 15N showed up in the three main pools we measured (in leachate vs. pore water in the soil column vs. soil). This is depicted in Figure 2, with most of the tracer (i.e. close to 100%) ending up in the three measured pools for the mid-layer legacy sediment and the relict A horizon. That we only saw ~60% recovery of the added tracer in these three pools in the surface legacy sediment horizon suggests that there are other important pools that may retain or lose 15N that we did not measure. We applied the term 15N retention to that still being held in the soil columns (i.e. 15N in the pore water and the soil). We believe both figures are needed. Figure 2 shows that for the deeper soil horizons we accurately measured the three main pools through which 15N moves, with much of the 15N we added initially lost in leachate prior to drought. Figure 3 shows that added 15N that stays in the soil columns can be greatly reduced when the soil experiences a drought-rewetting event. We incorporated some of this explanation into the text to better articulate the difference between 15N recovery vs. 15N retention and to justify the use of both figures.*

Results and discussion:
P11 L23: please spell out "atm%"

*Response: We added the definition for atom percent (and its shortened form of atom %) to the text, as suggested by Reviewer #1, as well, and so adjusted the wording to atom % at this and all other instances the text.*

P13 L9: If it is proposed that low NO3 uptake is the reason for the large NO3 leaching losses, please discuss some of the processes which may be governing NO3 uptake, and why these are low in this soil - . . . how does this compare with other milldams or equivalent textured soils?

*Response: We discussed some processes that may explain low $NO_3^-$ uptake in the soil, such as kinetic N saturation or preferential uptake of $NH_4^+$ over $NO_3^-$. We do agree, however, that comparing and contrasting the occurrences of these processes in similar soils is very useful, and added such a literature analysis to the discussion.*

P14 L24: I'm not familiar with the term "well-sorted soil".

*Response: Sorting describes the distribution of grain sizes within the soil. A well-sorted soil is composed of grains that are similar in size. We added this explanation to the text: "…a well-sorted soil, composed of grains of similar size,…".*

Conclusions:
P15 L20: The comment regarding restoration of the site may lead to an initial decrease in NO3 retention capacity – some comments around the magnitude and importance of this proposed decrease would be of value – how does it rate compared to the landscape as a whole?

*Response: As suggested in the general comments above, we added more discussion about the potential environmental benefits of restoration related to expected magnitude changes in $NO_3^-$ levels over the long-term, as it relates to the stream banks and the surrounding landscape as a whole.*

References:
-Castellano and Kaye (2009) not mentioned in paper

- Merrits et al (2010) not mentioned in paper.

*Response: The Castellano and Kaye (2009) reference was a remnant of previous edits to the manuscript and its removal from the references list must have been overlooked. We removed it from the current references list. The "Merritts et al (2010)" article, however, was cited in a note in Table 2, so it was left in the references list.*